# Complex Actions of FKBP12 on RyR1 Ion Channel Activity Consistent with Negative Co-Operativity in FKBP12 Binding to the RyR1 Tetramer

**DOI:** 10.3390/cells14030157

**Published:** 2025-01-21

**Authors:** Spencer J. Richardson, Chris G. Thekkedam, Marco G. Casarotto, Nicole A. Beard, Angela F. Dulhunty

**Affiliations:** 1CHEM1—Biotechnology, IP Australia, Woden Valley, ACT 2606, Australia; spencer.sichardson@ipaustralia.gov.au; 2Developmental and Regeneration Biology Laboratory, Victor Chang Cardiac Research Institute, 405 Liverpool St, Darlinghurst, NSW 2010, Australia; c.thekkedam@victorchang.edu.au; 3Biomolecular Interactions Group, Research School of Biology, Australian National University, Canberra, ACT 2601, Australia; marco.casarotto@anu.edu.au; 4Muscle Proteomics Group, Centre for Research in Therapeutic Solutions, University of Canberra, Bruce, ACT 2617, Australia; nicole.beard1971@gmail.com; 5Muscle Research Group, Eccles Institute of Neuroscience, John Curtin School of Mecical Research, Australian National University, Canberra, ACT 2601, Australia

**Keywords:** ryanodine receptor, FKBP12, ion channel activation, ion channel inhibition, skeletal muscle, myopathy

## Abstract

The association of the 12 KDa FK506 binding protein (FKBP12) with ryanodine receptor type 1 (RyR1) in skeletal muscle is thought to suppress RyR1 channel opening and contribute to healthy muscle function. The strongest evidence for this role is increased RyR1 channel activity following FKBP12 dissociation. However, the corollary that channel activity will decrease when FKBP12 is added back to FKBP12-depleted RyR1 is not well established, and when reported, the time- and concentration-dependence of inhibition vary over orders of magnitude. Here, we address this problem with an investigation of the molecular mechanisms of the FKBP12 regulation of RyR1. Muscle processing to obtain sarcoplasmic reticulum (SR) vesicle preparations enriched in RyR1 resulted in substantial FKBP12 dissociation from RyR1, indicating low-affinity binding. Conversely, high-affinity binding was indicated by some FKBP12 remaining bound to RyR1 after solubilization. We report, for the first time, an increase in the activity of FKBP12-depleted channels after the addition of exogenous FKBP12 (5 nM to 5 µM), followed by a reduction in activity consistent with inhibition after 20–30 min exposure to higher [FKBP12]s. Both the increase and later decline in activity were time- and concentration-dependent. The results suggest a high-affinity activation when FKBP12 binding sites on the RyR1 tetramer are partially occupied by FKBP12 and lower affinity inhibition as more RyR1 monomers become occupied. These novel results imply negative cooperativity in FKBP12 binding to RyR1 and a dynamic role for FKBP12/RyR1 interactions in intact muscle fibers.

## 1. Introduction

Ryanodine receptors (RyRs) belong to a class of calcium ion channel proteins that are found exclusively in the membrane of intracellular organelles, particularly the sarcoplasmic reticulum (SR) of skeletal and cardiac muscle. The skeletal (RyR1) and cardiac (RyR2) isoforms release the Ca^2+^ ions required for excitation–contraction coupling (ECC), where surface membrane depolarization initiates muscle contraction. ECC in skeletal muscle depends on a cascade of physical interactions between proteins linking the surface membrane voltage sensor (Ca_V_1.1) with RyR1, in contrast to cardiac muscle, where RyR2 is activated by Ca^2+^ entry through Ca_V_1.2. The FK506 binding proteins FKBP12 and FKBP12.6 bind with high affinity to RyRs [1,2,3], with FKBP12 used to purify RyRs, particularly for cryo-electron microscopy [4,5,6]. The binding of FKBP12 and FKBP12.6 to RyRs has been strongly implicated in normal channel function and shown to be disrupted in both skeletal and cardiac myopathies [7,8].

FKBP12 in skeletal muscle is generally considered to maintain low resting RyR1 channel activity and low SR Ca^2+^ leak. This role is strongly supported by studies showing that removing FKBP12 from RyR1 increases channel open probability (*P_o_*) [9,10], which, in some reports, is associated with an increase in sub-conductance activity [11]. However, the reported effects of adding FKBP12 to RyR1 channels are not consistent with strong inhibition or the high-affinity binding of FKBP12 to RyR1. High concentrations of FKBP12 and long exposure periods are often required to see weak channel inhibition. In the most comprehensive study thus far, Mei et al. [10] reported inconsistent changes in *P_o_* during exposure to [FKBP12] of less than 5 µM for less than 30 min. They found that adding 5 µM of FKBP12 to RyR1 channels for 30 min did not reduce channel activity, unless the FKBP12 bound to the channels was experimentally removed prior to adding FKBP12, with a greater decrease in the presence of S107. They reported a progressive increase in FKBP12 binding when FKBP12-depleted RyR1 was incubated with 1 µM of FKBP12 for up to 2 h. Barg et al. [9] reported that FKBP12-depleted RyR1 channels were inhibited when 5–10 µM of FKBP12 was added to the bilayer solution for 5–10 min. On the other hand, Venturi et al. [12] describe a 75% decline in RyR1 activity within a few seconds of adding 500 nM FKBP12 to RyR1 channels, suggesting a higher-affinity binding.

The broad aim of the present study was to better understand the molecular mechanisms of the FKBP12 regulation of RyR1 channels and the likely physiological regulation of the channels by FKBP12 in skeletal muscle. We have examined the relative amounts of FKBP12 that remain associated with RyR1 following SR processing and the extent to which this partial depletion in FKBP12 might alter the effect of exogenous FKBP12 binding on RyR1 channel gating. This study is particularly relevant given the differing reports of the effects of adding FKBP12 back to FKBP12-depleted RyR1 and the reported time- and concentration-dependence of exogenous FKBP12 binding to RyR1 reported by Mei et al. [10]. These questions are important in understanding how changes in FKBP12 binding to RyR1 might alter channel activity in the cellular environment. These questions also address a previously unexplored possibility that the affinity of the FKBP12 binding sites on each of the subunits of the RyR1 tetramers might depend on the occupation of the sites, as suggested for RyR2 [13].

## 2. Methods

### 2.1. Preparation of Rabbit Skeletal Muscle SR Vesicles and RyR1 Solubilization

SR vesicles were prepared from rabbit back and leg muscles, and RyR1-enriched vesicles were fractionated using discontinuous sucrose gradient techniques (Figure 1A). Native rabbit RyR1 channels are routinely studied using the band 3 (B3) or band 4 (B4) vesicles [11,14,15]. In our experience, there is no difference between any aspect of the ion channel activity between the two fractions. The native RyR1 protein in sucrose gradient SR vesicles remains associated with the SR membrane and many of the proteins that bind to RyR1, including FKBP12, calsequestrin, triadin, and junctin [16,17]. Only B4 vesicles were used in the following Co-IP experiments.

RyR1 was partially purified from B4 vesicles, which were homogenized for 1 min every 10 min in a Potter homogenizer on ice for 30–60 min in solubilization buffer containing (mM): 25 PIPES, 1000 NaCl, 1 DTT, 0.1 EGTA, 0.92 CaCl_2_, and 0.5 AMP; 0.5% CHAPS/5% L-α-phosphatidylcholine, with EDTA-free complete protease inhibitor, at pH 7.4. Then, insoluble membrane fragments were removed by 20 min centrifugation at 163,200× *g*. The supernatant was loaded onto a continuous 5–20% sucrose gradient (5–20% sucrose in solubilization buffer) and centrifuged overnight at 71,935× *g* for 14–16 h. Gradient fractions showing significant amounts of protein at ~500 kDa on silver-stained SDS-PAGE gels were pooled, concentrated to ~1 mg/mL using a Vivaspin–2 concentrator, and stored at −70 °C. The presence of RyR1 in these fractions was confirmed with Western blot, and the RyR1 concentration was determined using a DC Plus protein assay kit (supplied by Bio-Rad Laboratories Pty Ltd Unit 1A, 62 Ferndell Street, South Granville, NSW 214, Australia. Western blots of the purified fractions confirmed that there was no detectable CSQ, junctin, or triadin remaining associated with RyR1.

### 2.2. FKBP12 Expression and Purification

GST-FKBP12 was expressed in *E. coli* in a pGEX2TK vector and purified by glutathione-agarose affinity chromatography [18]. GST-FKBP12 or GST-free FKBP12 were purified by FPLC following thrombin cleavage [13,15].

### 2.3. SDS-PAGE and Western Blotting

Vesicle proteins were separated by SDS-PAGE, and protein bands were stained with Coomassie blue, silver-stained, or transferred to a PVDF membrane. Non-specific antibody binding was blocked with skim milk powder or Bovine Serum Albumin (BSA) in Phosphate-Buffered Saline (PBS) containing the following (mM): 137 NaCl; 7 Na_2_HPO_4_; 2.5 NaH_2_PO_4_·H_2_O; and 2 EGTA (ethylene glycol-bis[b-aminoethyl ether]N,N,N′,N′-tetraacetic acid), pH adjusted to 7.4. The anti-RyR antibody was 34C (RyR1/RyR2 antibody, Development Studies Hybridoma Bank, Iowa City, IA, USA) was diluted to 1:6000 in TPBS (0.05% Tween-20 PBS). The FKBP antibody H5 (α-FKBP12/FKBP12.6 antibody, (Santa Cruz Biotechnology, Inc., 10410 Finnell Street, Dallas, TX 75220. USA) detects FKBP12 and FKBP12.6 and was diluted to 1:1000 in TPBS. The monoclonal VIIID12 anti-CSQ1 antibody was from Abcam (Discovery Drive, Cambridge Biomedical Campus, Cambridge, CB2 0AX, UK), 1:2000. The monoclonal anti-triadin antibody (IIG12) was from Sigma-Aldrich (Unit 2, 10 Anella Ave, Castle Hill, NSW 2154, Australia), 1:1000. The anti-junctin antibody was against a junctin peptide (KLH-C-SKHTHSAKGNNQKRKN-OH; GL Biochem, Shanghai, China) supplied by IMVS Veterinary Services, Australia, 1:1000. The second anti-mouse IgG-conjugated HRP antibodies were from Santa Cruz (Santa Cruz Biotechnology, Inc., 10410 Finnell Street, Dallas, TX 75220, USA) 1:6000.

### 2.4. Co-Immunoprecipitation (Co-IP) and Densitometry

SR vesicles were incubated without (control) and with FKBP12 or GST-FKBP12 for 1 h at 37 °C, as specified in the Results (Section 3), and then subjected to anti-RyR Co-IP [13,15], followed by SDS-PAGE and Western blot. Western blots were scanned and band density quantified using Bio-Rad ImageLab (Bio-Rad Laboratories Pty Ltd Unit 1A, 62 Ferndell Street, South Granville, NSW 214, Australia) and then background density was subtracted. An approximately linear relationship between band density and [RyR1] or [FKBP12] was established for 34C and H5 [13,15] and is apparent in most gels analyzed for the present study (Appendix A). Unless stated otherwise, the data for each gel are expressed as a ratio of FKBP/RyR1 following FKBP12 addition or further vesicle processing relative to the ratio of FKBP/RyR1 before FKBP12 addition or further vesicle processing. Since the relative densities are determined for bands on the same gel, factors such as image exposure time and other factors, including antibody affinity, are effectively eliminated. The data reveal changes that occur due to FKBP12 addition or further vesicle processing, but the analysis does not reveal absolute amounts of either RyR2 or FKBP12.

### 2.5. Lipid Bilayers and RyR2 Channel Incorporation

Lipid bilayers were formed from a mixture of phosphatidylethanolamine, phosphatidylserine, and phosphatidylcholine (in n-decane), as previously described [11,14]. The SR vesicles were added to the cis solution and incorporated into bilayers with their cytoplasmic surface facing the cis chamber. RyR1 channel activity was recorded with symmetrical 250 mM Cs^+^ in the cis and trans chambers, with physiological diastolic Ca^2+^ concentrations of 1 µM Ca^2+^ in the cis chamber (unless otherwise stated) and 1 mM Ca^2+^ in the trans chamber.

### 2.6. Single Channel Recording and Analysis

Electrodes in the cis and trans solutions detected current flow through RyR1 channels, and voltage clamped the potential across the bilayer (Vcis-Vtrans), which was switched between −40, 0, and +40 mV every 30 s throughout the experiment [11,14]. Channel currents were analyzed over 3 consecutive 30 s periods (to a total of approximately 80 s) at each potential before exposure to FKBP12 and then at 4 to 8 min intervals following FKBP12 addition to the cytoplasmic (cis) bilayer solution. When only one channel was active in a bilayer, *P_o_*, mean open time (*T_o_*) and mean closed time (*T_c_*) were measured using the in-house programs Channel 2 (developed by P. W. Gage and M. Smith, John Curtin School of Medical Research) or Channel 3 (developed by N. W. Laver, University of Newcastle), with open channel discriminators set above the baseline noise at ~20% of the maximum single channel conductance. When 2–4 channels were active, *P_o_* was approximated from the mean current, *I*′, as outlined in the Results [13]. Channel activity was analyzed before FKBP12 addition and then after adding FKBP12 at various times from 0.2 min up to 30 min post-addition, as specified in the Results.

Because RyR channel activity characteristically varies over a wide range between channels [19], parameter values during exposure to FKBP12 are also expressed relative to the internal control values for each individual channel to give equal weighting to the effect of FKBP12 channels with high and low activity [13]. The average relative data presented in the manuscript are the average of the relative values calculated for each individual bilayer experiment.

### 2.7. Statistics

Data are presented as mean ± s.e.m. Significance was evaluated using Student’s *t*-test. For all Co-IP and channel data, the threshold for significance was *p* < 0.05. For the Co-IP data, unless otherwise stated, the analysis was 2-tailed, and the type was “type 1” (paired). For analysis of ion channel data, the analysis was either 1-tailed or 2-tailed and either type 1, type 2, or type 3 as appropriate. The effects of FKBP12 on channel activity were not voltage-dependent, so measurements at +40 and −40 mV were included in the average data, as the two voltages provide independent measures of the effects of FKBP12 at each concentration with current flow through the pore in opposite directions [13].

## 3. Results

### 3.1. CoIP Analysis of FKBP12 Association with RyR1

#### 3.1.1. FKBP12 Association with RyR1 in P2 and B4 Vesicles

FKBP12 associated with RyR1 in B4 vesicles was first compared with that in the muscle homogenate (Figure 1B). The FKBP12 band density was expressed relative to the density of the RyR1 band in the same lane, and then, the FKBP12/RyR1 ratio in B4 was normalized to the FKBP12/RyR1 ratio in the homogenate in the same gel. There was a significant 50% loss of FKBP12 from RyR1 during processing from the homogenate to B4 (Figure 1B). In contrast to the homogenate and B4, there was only a small although significant decline in average relative FKBP12/RyR1 in B4 compared to P2 (Figure 1C). The difference in the average FKBP12/RyR1 ratio between P2 and B4 was significantly less than the difference between the homogenate and B4. This result implies that most of the dissociation of FKBP12 from RyR1 occurs during the initial processing of the homogenate rather than during the following sucrose gradient to enrich the samples with vesicles containing RyR1. In contrast to RyR1, very little FKBP12 was lost from RyR2 during initial homogenate processing [13]. Since the processing techniques for both skeletal and cardiac muscle homogenate were identical, FKBP12 was apparently more easily dissociated from RyR1 than from RyR2 at this stage of processing, possibly suggesting a lower-affinity binding of FKBP12 to RyR1 in the RyR1 homogenate.

**Figure 1 cells-14-00157-f001:**
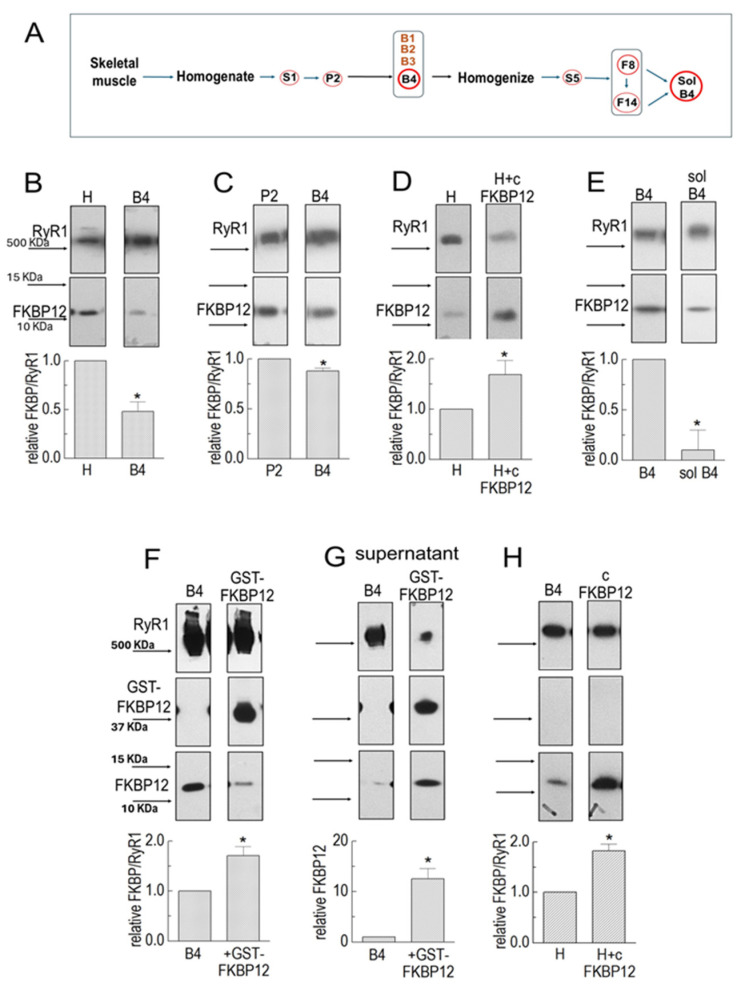
FKBP12 dissociation from RyR1 during SR processing. (**A**) Sequential centrifugation and sucrose gradient fractionation in isolating SR vesicles enriched in RyR1 channels and solubilized RyR1. Relevant solute fractions are labeled S1 and S5, the relevant pellet is labeled P2, and sucrose gradient bands are labeled B1 to B4. (**B**–**H**) Co-IP of RyR1 and FKBP12 in SR preparations. Co-IP samples were loaded at protein concentrations of approximately 3, 6, and 9 μg. (**B**) RyR1 and FKBP12 in homogenate (H) and B4 vesicles (*n* = 5). (**C**) RyR1 and FKBP12 in P2 and B4 vesicles (*n* = 6). (**D**) RyR1 and FKBP12 in control-incubated homogenate and homogenate incubated with GST-cleaved FKBP12 (cFKBP12) (*n* = 3). (**E**) RyR1 and FKBP12 in B4 and solubilized B4 (*n* = 5). (**F**) RyR1, GST-FKBP12, and FKBP12 in incubated B4 without GST-FKBP12 (control) and B4 incubated with GST-FKBP12 (*n* = 4). (**G**) Supernatants following Co-IP control-incubated B4 and B4 incubated with GST-FKBP12 (*n* = 5). (**H**) B4 and B4 incubated with GST-cleaved FKBP12 (cFKBP12) (*n* = 3). (**B**–**H**) The images in each figure show immune-stained bands from the same SDS-PAGE gel. The upper image in each panel shows the RyR1 band, and the lower image shows the FKBP12 band. An additional band is shown for GST-GKBP12 in (**F**,**G**). All vertically aligned images in one panel were obtained from the same lane. MW markers are labeled in (**B**,**F**), and corresponding arrows indicating marker positions are shown in all panels. Unless otherwise stated, the graphs in each figure show the average relative FKBP12/RyR1 ratios. The ratios were first calculated for RyR and FKBP bands in the same lanes, and then, the ratio for the right-hand lane was expressed relative to ratios for the left-hand “control” lane in the same blot. The average values are indicated by broad vertical bars, and the s.e.m. is indicated by the vertical capped lines. The *n* values refer to the number of individual experiments. The total FKBP12 density in (**F**) was calculated as (FKBP12 + GST-FKBP12/2). The asterisks * in (**B**–**H**) indicate significant differences from the control, left hand bars.

The degree of RyR1 saturation with FKBP12 in the homogenate was determined by incubating the homogenate with 10 µM of FKBP12 for 1 h before co-immunoprecipitation. There was a significant 1.69 ± 0.28-fold increase in FKBP12 bound to the RyR1 monomers (Figure 1D). The increase in FKBP12 associated with RyR1 shows that, even at this early stage of processing, RyR1 in the homogenate was less than saturated with FKBP12.

FKBP12 remained associated with RyR1 in solubilized B4 preparations in detectable amounts, with an average significant reduction to 90% below that in the B4 vesicles (Figure 1E), although as much as 30% remained bound in some preparations. Small amounts remaining bound to solubilized RyR1 were not surprising because FKBP12 has been visualized as bound to RyR1 in cryo-EM images in protein purified by means other than FKBP12 pull-down [20].

#### 3.1.2. Does Exogenous GST-FKBP Displace Endogenous FKBP?

The experiments in Figure 1D,E show that the total FKBP12 bound to RyR1 but do not distinguish between exogenous and endogenous FKBP12 binding, whether the exogenous protein binds only to unoccupied sites, or whether it exchanges with endogenous protein. The higher-molecular-mass GST-FKBP12 fusion protein overcomes this problem and enables a distinction between exogenous and endogenous FKBP12 (Figure 1F). In calculating extra FKBP12 binding, it was necessary to account for the dimerization of GST-FKBP12 [13] so that two FKBP12 molecules were associated with each GST-FKBP12 bound to RyR1. Therefore, the measured density of the GST-FKBP12 band was divided by a factor of two before calculating the ratio of GST-FKBP12/RyR1.

There was an average 75 ± 18% increase in the total FKBP12 (endogenous + exogenous) bound to RyR1 following incubation with GST-FKBP12 (Figure 1F). It is notable that the density of the endogenous 12 KDa FKBP12 band was substantially reduced after exposure to GST-FKBP12, indicating that the exogenous fusion protein displaced some endogenous protein, as previously reported [1], and suggesting low-affinity binding. This conclusion was confirmed by the appearance of FKBP12 in the supernatant following the precipitation of the GST-FKBP12-RyR1 complex (Figure 1G). The control-incubated supernatant contained some FKBP12, presumably also bound to the RyR1 in the supernatant. The substantial increase in FKBP12 after incubation with GST-FKBP12 confirms that exogenous GST-FKBP12 is exchanged with endogenous FKBP12 during incubation. A possible effect of the GST tag on the binding of GST-FKBP12 to RyR1 was explored using FKBP12 cleaved of the GST tag (Figure 1H) [21]. The 82.5 ± 13% increase in cleaved FKBP12 binding was not significantly different from the 75 ± 18% increase seen with GST-FBP12.6.

#### 3.1.3. S107 Stabilizes FKBP12 Binding to RyR1

The drug S107 stabilizes the interaction between FKBP12 and RyR1 [10]. S107 was added to all solutions used for homogenizing and processing the muscle. There was no significant loss of FKBP12 between the homogenate and P2 or between P2 and P4 in the presence of S107 (Figure 2A–C). The substantial loss of FKBP12 between the homogenate and P2 was significantly reduced in the presence of the drug, while the smaller loss between P2 and B4 was unaffected (Figure 2B,C).

### 3.2. Electrophysiological Analysis of FKBP12 Association with RyR1 

FKBP12 was added to native RyR1 channels in the B3 and B4 vesicles and partially purified RyR1 from solubilized B4 vesicle preparations. For most of these experiments, the GST tag was cleaved from FKBP12 to remove any possibility of GST binding to RyR1 and independently modifying activity [21], although there was no apparent difference in pilot experiments between the effects of the cleaved FKBP12 and GST-FKBP12 on RyR1 activity. Parameter values were measured in the presence of added FKBP12 and normalized to values prior to exposure to FKBP12 for each individual channel [13]. The absolute values are shown to illustrate the range of *P_o_* values, the duration of open events, and the duration of the intervening closed periods. Relative values are also shown because *P_o_*, in particular, can vary between RyRs from <0.00001 to 1.0, so the average *P_o_* is biased toward the highest-activity channels [13]. Normalization removes this bias so that the average relative data reflect the overall effect of FKBP12 on all individual channels. Both relative and absolute parameter values are shown in Figure 3, Figure 4, Figure 5, Figure 6 and Figure 7.

#### 3.2.1. FKBP12 Added to Native RyR1 Channels

In initial experiments, 1 nM of FKBP12 was added to the cytoplasmic solution for ~5 min and then increased to 5 nM by adding a further 4 nM of FKBP12. The experiments were performed with low-activity channels in the presence of 1 µM cytoplasmic Ca^2+^, with channels activated by 10 µM of cytoplasmic Ca^2+^ and with rapidly inactivating channels in the presence of 10 µM of Ca^2+^ plus 2 mM of ATP in the cytoplasmic solutions (Figure 3A,B). Luminal Ca^2+^ was 1 mM in all cases. *P_o_* data were obtained for single-channel recordings with only one channel opening in the bilayer. Multiple channel recordings were obtained in some cases with higher-activity channels exposed to 10 µM of cytoplasmic Ca^2+^ both with ATP and without ATP. Channel activity in multiple channel recordings was measured as the fractional mean current, *I*′*F*, which approximates the average *P_o_* of the two to four channels opening in the bilayer [13]. When single-channel recordings were obtained, as well as multiple-channel recordings, with 10 µM of cytoplasmic [Ca^2+^] and 10 µM of Ca^2+^+ 2 mM ATP, the average effects of FKBP12 were calculated from the combined *P_o_* and *I*′*F* values (Figure 3B, center and right).

No consistent changes in relative *P_o_* or relative *I*′*F* were apparent within the first 10 min of exposure to 1–5 nM of FKBP12 under any condition (Figure 3A,B). However, there was a significant 2- to 3-fold increase in relative activity after 15 min of exposure to 5 nM of FKBP12 with 1 µM of cytoplasmic Ca^2+^. All subsequent experiments were performed with 1 µM of cytoplasmic Ca^2+^. Exposure to 0.2 µM of FKBP12 produced significant 1.5- to 2-fold increases in the average relative *P_o_* after 14 and 20 min, with a trend toward a progressively greater activity at longer times (Figure 3A,C). Increasing the [FKBP12] to 1.0 µM resulted in significant increases in the average relative *P_o_* after <5 min, with activity increasing ~3-fold after 5 to 10 min. This was followed by a trend toward a decline in activity (i.e., less activation) with longer exposure (Figure 3D). One experiment was performed with 5 µM of FKBP12, and both the *P_o_* and relative *P_o_* values for this channel are included as open red circles with the average data for 1 µM of FKBP12. The relative *P_o_* with 1 µM and 5 µM of FKBP12 declined to less than control levels at positive potentials in three of the six channels but did not fall below control levels in any channels at negative potentials. The biphasic effects of FKBP12 and concentration dependence are more clearly illustrated in Figure 3E, where the data in Figure 3B–D have been collected into three groups (1–5 min, 7–10 min, and 11–20 min) after exposure to FKBP12. The time-dependence of activation can be clearly seen with 5 nM and 0.2 µM of FKBP12. The decline in activity apparent with exposure to 1.0 µM FKBP12 is reminiscent of the effects of FKBP12 on RyR2 [13].

#### 3.2.2. Single-Channel Gating Parameters During Exposure to FKBP12

Gating parameters provide insight into dynamic changes within the channel that occurred with FKBP12 binding to RyR1. The changes in gating parameters following exposure to FKBP12 presented a complex picture, suggesting that there were complex changes occurring within the RyR1 protein (Figure 4). The most striking effects were seen with exposure to 1–5 nM of FKBP12, where the closed durations were not altered but the significantly longer open times (Figure 4A, left) were consistent with the slow increase in *P_o_* (Figure 3 above). In marked contrast, the channel open times were not consistently altered by exposure to 0.2 µM to 5 µM of FKBP12 (Figure 4B,C, left), while the biphasic changes in mean closed times (Figure 4B,C, center) could account changes in *P_o_*. During early exposure to 1 µM and 5 µM of FKBP12, the briefer closures contributed to the increased *P_o_* (and might be a result of the longer openings). The reversal of this decline, with closed durations becoming longer after longer exposures to 1 µM and 5 µM of FKBP12 (Figure 4B,C, center), underpins the decreasing *P_o_* at longer times (Figure 3 above).

#### 3.2.3. FKBP12 Added to RyR1 Channels in Solubilized SR

The partially purified solubilized RyR1 channels were more active before FKBP12 addition than native RyR1 channels, with at least two channels opening simultaneously in the bilayer (Figure 5A–D, left column). Therefore, the activity of all solubilized channels is expressed as *I*′*F*, determined by dividing the mean current (*Imean)* by the maximum current (*Imax)*. The average control *I*′*F* values, varying from 0.09 and 0.2 (Figure 5A(iv)–D(iv)), were higher than the average control *P_o_* values for the native B3/B4. The relative *I*′*F* is shown to compare the effect of FKBP12 in all bilayers, irrespective of their initial activity (Figure 5A(ii,iii)–D(ii,iii)).

The solubilized RyR1 channels responded more rapidly to FKBP12 than the native RyR1 channels. The average *I*′*F* and relative *I*′*F* values tended to increase within the first 1–3 min following the addition of 1–10 nM of FKBP12, with a significant increase in the relative *I*′*F* after 5 min (Figure 5A). The average relative *I*′*F* reached significance within 1–3 min of adding 200 nM to 5 µM of FKBP12 (Figure 5B(iii)–D(iii)). There was a trend toward a higher average *I*′*F* and relative *I*′*F* as the FKBP12 concentration increased up to 1 µM and then a significantly smaller initial increase with 5 µM of FKBP2 (Figure 5D). The data in Figure 5A–D have been collected into 0–5 min, 6–9 min, and 10–35 min groups in Figure 5E. There were significant increases in relative *I*′*F* in all time groups at all FKBP12 concentrations. The relative *I*′*F* with 1 µM of FKBP12 was significantly greater than the relative *I*′*F* with 1–10 nM of FKBP12 in all time groups and significantly greater than with 5 µM of FKBP12 in the two shorter time groups.

#### 3.2.4. S107 Addition to Cytoplasmic Solution Containing FKBP12

The co-IP results show that the addition of S107 to the homogenization and processing solutions prevented the loss of endogenous FKBP12. When 20 µM of S107 was added to the cytoplasmic bilayer solution following the addition of 0.2 µM of FKBP12, the activity of both native RyR1 and solubilized RyR1 eventually fell to values that were significantly lower than the control activity (Figure 6A,B). The heightened activity after FKBP12 addition alone was maintained or slightly increased for several minutes after adding S107 before activity declined over the next 20 min to values that were significantly lower than those in the presence of FKBP12 alone and significantly lower than the control activity. It is possible that activity may have declined after these very long exposures to 200 nM of FKBP12 in the absence of S107, although a trend toward a decline in the presence of S107 was apparent after 16–20 min that was not apparent in the absence of the drug. The decline in activity with S107 is consistent with the drug increasing RyR1 affinity for FKBP12 and, therefore, allowing sufficient binding with 0.2 mM of FKBP12 to overcome activation and reveal a significant inhibitory effect on channel activity, as reported by Mei et al. [10].

To summarize, activity increased sooner after adding FKBP12 to the solubilized channels than in the native channels, although the maximum increases in activity were around 2-fold to 3-fold in both preparations. The biphasic effect of FKBP12 seen in the native channels was less apparent in solubilized channels, although the maximum increase in activity during exposure to 5 µM FKBP12 in both cases was less than the maximum increases with 1 µM of FKBP12.

#### 3.2.5. Insights into the Effects of FKBP12 Binding to RyR1 on Ca^2+^ Release from the SR

While *I*′*F* reflects the average *P_o_* of channels in the bilayer, information is lost when *Imean* is divided by *Imax* to calculate *I*′*F. Imean* is the current crossing the bilayer and predicts Ca^2+^ efflux from the SR in the absence of soluble modulators present in the cytoplasm. *Imean* increased significantly with all [FKBP12]s, with a maximum of ~3-fold with 1.0 µM of FKBP12 (Figure 6C(i–iv)). *Imax*, reflecting the number of channels opening simultaneously, showed a small but significant 1.2-fold increase with 1.0 and 5.0 µM of FKBP12 (Figure 6D(i–iv)). Similar trends are apparent in the relative *Imax* and *Imean* data (Figure 7), although with reduced variability. Therefore, the increased activity with FKBP12 was due to an increase in the activity of individual channels present in the bilayer, rather than the number of channels opening. An increase in the number of channels opening would be reflected in step changes to multiples of the maximum single-channel conductance (~15 pA), as seen with coupled gating [22]. Step increases in current are not apparent in the data but rather a broad spectrum of *Imax* values, which reflects a contribution of sub-conductance openings [11] before and after the addition of FKBP12.

A significant decline in activity during exposure to 0.2 µM of FKBP12 was seen when S107 was added shortly after FKBP12 (Figure 7E), with relative *Imax* and *Imean* reduced to levels significantly lower than control. The decline in *Imax* could be due either to a decrease in the number of channels opening or to channels opening to fewer and/or lower sub-conductance levels [10,11]. Either scenario is consistent with the inhibition of RyR1 by FKBP12 and with reduced Ca^2+^ leak from the SR.

## 4. Discussion

In summary, we show a progressive decline in FKBP12 associated with RyR1 during the handling of skeletal muscle to obtain SR vesicles enriched in RyR1 channels, although we also show measurable amounts of FKBP12 remained associated with RyR1 following further processing during SR vesicle solubilization. FKBP12 added back to partially FKBP12-depleted channels bound to unoccupied sites on RyR1 and exchanged with a fraction of the remaining bound FKBP12. The drug S107 prevented the loss of FKBP12 during processing. FKBP12’s association with FKBP12-depleted RyR1 channels initially caused time-dependent and concentration-dependent increases in channel activity. Channel activity then declined during longer exposure to FKBP12 and fell below control levels in the presence of S107. We interpret the initial increase in channel activity as channel activation and the decline in activity as channel inhibition (Figure 8). The changes in channel activity are consistent with a model of negative co-operativity in FKBP12 binding to RyR1, as described previously with FKBP12 binding to RyR2 [13].

### 4.1. Biphasic Changes in RyR1 Channel Activity with FKBP12 Binding 

This is the first report of an increase in RyR1 activity associated with FKBP12 binding. The general assumption that FKBP12 inhibits RyR1 channels is based on an increase in channel activity consistently seen when FKBP12 is removed from RyR1 [9,10,11,14]. The evidence from adding FKBP12 to FKBP12-depleted RyR1 is less strong and generally not consistent with the assumption of high-affinity binding. A weak decline in channel activity has been reported previously but only after FKBP12-depleted channels were exposed to 5 µM of FKBP12 for 30 min [10] or 5 µM of FKBP12 for >5 min [9]. Similarly, we can see a decline in channel activity from the activated level after 15 to 30 min of exposure to 0.2 to 5 µM of FKBP12 (Figure 4, Figure 5, Figure 6 and Figure 7). A decline in activity only after long exposures to high concentrations of FKBP12 in the absence of S107 is more consistent with low-affinity binding of FKBP12 to RyR1 rather than the high-affinity binding indicated by previous biochemical studies [1,3].

Although we are unable to find previous reports of RyR1 activation preceding inhibition during exposure to FKBP12, our approach differed from previous studies. In particular, we examined the effects of adding FKBP12 to native channels containing up to 50% endogenous FKBP12, as well as to solubilized channels, which contained small but significant amounts of endogenous FKBP12. In addition, we analyzed *P_o_* continuously following exposure to added FKBP12. Mei et al. [10] found no significant change in *P_o_* after 30 min of exposure to 5 µM of FKBP12 unless channels were pre-treated to remove FKBP12. Notably, they mention that there were less consistent changes with shorter exposure times and lower concentrations of FKBP12. Indeed, we can see fewer significant changes in the raw *P_o_* data (Figure 3C(iii),D(iii)), where average values are dominated by the highest-activity channels, although there were consistent trends toward an increase in *P_o_*. Significant increases were unmasked in the relative data, suggesting that low-activity channels may be more sensitive to FKBP12 than high-activity channels. Barg et al. [9] reported a decline in activity after FKBP-stripped RyR1 channels were exposed to 5–10 µM of FKBP12 for 5 min but did not address channel activity with briefer exposure and lower [FKBP12].

The decline in activity following activation was less apparent in solubilized RyR1, possibly because the initial binding of added FKBP12 was to high-affinity excitatory sites. The increase in activity with 5 µM of FKBP12 was less than with 0.2 or 1.0 µM of FKBP12, and there was a strong trend toward reduced activation with 1.0 and 5 µM of FKBP12 compared with 5 nM and 0.2 µM of FKBP12. These results suggest that activation with the higher [FKBP12] was overwhelmed by the onset of inhibition. The activation associated with FKBP12 binding to RyR1 provides new information on FKBP12 interactions with the RyR1 tetramer. The presence of activation raises the possibility that both the activation and the relief of inhibition contribute to the increased activity when FKBP12 dissociates from RyR1 following treatment with FK506 or rapamycin [9,10,11,14]. Notably, although a substantial fraction of bound FKBP12 is dissociated from RyR1 by FK506 or rapamycin, an amount equivalent to that bound to solubilized RyR1 remains [15]. Therefore, RyR1 activity might be expected to increase to activated levels as amounts of FKBP12 bound to the RyR1 tetramer are reduced into the activation concentration range.

The changes in single-channel gating indicate that activation depends on an increase in open times, while inhibition is governed by the stabilization of the closed state. Both activation and inhibition in solubilized channels were dominated by changes in the relative *Imean*, implying that the activity of channels that were open in the bilayer before the addition of FKBP12 was altered by FKBP12 binding rather than changes in the number of channels opening. The one exception was the strong inhibition in the presence of S107 with *Imax* and *Imean* similarly reduced, consistent with an increase in sub-conductance activity [11], to the extent that all channels failed to open to their full conductance. These results again suggest that separate mechanisms underlie the activation and inhibition phases of FKBP12 binding to RyR1.

The ability of S107 and its precursor JTV519 to bind to RyRs and enhance FKBP12/12.6 binding is well documented [10,23,24,25]. S107 prevented the loss of FKBP12 from RyR1 between the homogenate and P2 fractions (Figure 2A,B). We also show that the FKBP12-induced decline in relative *P_o_* and *I*′*F* was enhanced by S107. We cannot exclude the possibility that longer exposures to FKBP12 in the S107 experiments contributed to the greater decline in *P_o_*. However, activity declined shortly after adding S107 before any decline in the absence of S107 (Figure 6A,B), suggesting a contribution from S107, as previously reported [10].

### 4.2. Occupation of FKBP12 Binding Sites on RyR1 Channels

The occupation of FKBP12 binding sites in RyR1 can be estimated assuming that RyR1 channels contain four FKBP12 binding sites, one site on each subunit of the tetramer [4,5,6,20,26]. The 1.69-fold increase in FKBP12 binding following homogenate saturation with cFKBP12 indicates 2.4 FKBPs per RyR1 (4/1.69 = 2.37) in the homogenate. RyR1 occupation by FKBP12 in B4 vesicles was calculated from the results of three different experiments: (1) the 1.83-fold increase in GST-cleaved cFKBP12 binding suggests 2.2 FKBP2/RyR1 (4/1.83 = 2.2); (2) the 1.71-fold increase after saturation with GST-FKBP12 indicates 2.34 FKBP2/RyR (4/1.71 = 2.34); (3) the 52% lower FKBP12/RyR1 in B4 than in the homogenate (Figure 1B) suggests 1.25 FKBP12/RyR1 (2.4 × 0.52 = 1.25) in the B4 vesicles. Combined, these estimates suggest ~1.9 FKBP12/RyR1 in B4. The occupancy of solubilized RyR1 was ~10% of that in B4 (Figure 1E), so most solubilized RyR1 lacked any bound FKBP12. These are the first estimates of RyR1 occupancy based solely on measurements of RyR1 saturation using added FKBP12. Different approaches have yielded higher estimates of RyR1 occupancy in skeletal muscle homogenate: 96% in rabbit muscle based on a Kd value of 2.6 nM and a cytosolic [FKBP12] of 3 µM [1]; 95% in mouse muscle with a cytosolic [FKBP12] of 1.3–1.6 µM and an 8–30 nM affinity of FKBP12 for RyR1 [3]. The estimates would be lower if some FKBP12 binding sites with lower affinities were considered. Indeed, it has been suggested that a reduction in FKBP12 affinity from ~100 nM to >600 nM via PKA phosphorylation would reduce the amount of FKBP12 bound to RyR1 assuming a cellular [FKBP12] of 200 nM [8].

### 4.3. A Negative Co-Operativity Model for FKBP12 Binding to RyR1

Our hypothesis is that changes in RyR1 channel activity with increasing [FKBP12] and longer exposure times depend on conformational changes in the protein with the increasing occupancy of FKBP12 binding sites and flow-on changes in channel gating [13]. Our assumptions are that (i) RyR1 affinity for FKBP12 is reduced, as more subunits in the RyR1 tetramer are occupied by FKBP12, (ii) the highest-affinity binding is to non-adjacent subunits, (iii) binding to one subunit reduces the affinity for binding to its nearest neighbor subunits, and (iv) high-affinity binding activates RyR1 while lower-affinity binding inhibits the channel. FKBP12 bound to solubilized RyR1 is consistent with a population of FKBP binding with higher affinity and consistent with FKBP12 present in the CryoEM of RyR1 purified without GST-FKBP precipitation [20]. Conversely, a population of FKBP12 bound to RyR1 with lower affinity is consistent with the amounts of FKBP12 lost during the processing of SR vesicles. Microscale thermophoresis experiments indicate changes in affinity as the FKBP12/RyR1 ratio changes with inflections in the binding data at 10 nM of FKBP12 and 1.0 µM of FKBP12 [13]. These changes in affinity could reflect the activation that we report with FKBP12 concentrations around 10 nM and inhibition with FKBP12 concentrations around 1.0 µM. A reduction in affinity as more FKBP12 binds to the RyR1 tetramer implies negative co-operativity in this binding process.

Changes in RyR1 affinity for FKBP12 with increasing subunit occupation by FKBP12 could produce different effects of further FKBP12 binding to RyR1 channel gating because structural changes within each subunit and interactions between adjacent subunits can be transmitted through the protein to the channel gating region by a cascade of changes in residue orientations [27,28]. To explain the opposite effects that we see with increasing [FKBP12], we hypothesize the following: Residues in non-adjacent subunits are modified by FKBP12 binding in a manner that reduces the affinity of their nearest neighbors for FKBP12 and increases channel activity.FKBP12 binding with lower affinity to adjacent subunits alters inter-subunit interactions in a manner that decreases channel activity.The probability of adjacent subunits being occupied by FKBP increases as more subunits are occupied by FKBP, with higher concentrations of FKBP and longer exposure times.

The model (Figure 8) is feasible in the context of molecular rearrangements that have been described within the RyR1 tetramer under different conditions [27,28] and are thus possible upon FKBP12 binding to a single site on each of the four subunits. The FKBP12 binding site, located in the most mobile region of RyR1, is well defined at an atomic level [4,5,6,20]. Although located ~100 nM from the pore, mutations and ligand binding in the area disrupt channel gating via the knock-on effects of residue re-orientation along defined pathways throughout the protein to the gating regions [6,29]. It was suggested 25 years ago and more recently that greater mobility due to the “unzipping” of interdomain interactions could increase RyR1 activity and conversely that “zipping” interactions could reduce mobility and channel opening [6,20,28,30,31,32,33]. We suggest that the structural consequences of FKBP12 binding to one or two non-adjacent subunits alter their orientation to allow greater mobility within the tetramer and at the same time reduce the affinity of unbound subunits for FKBP12. Lower-affinity FKBP12 binding to adjacent subunits with increasing [FKBP12] and longer exposure times could stabilize inter-subunit interactions, reduce mobility within the corona, and consequently reduce channel activity. FKBP12 is thought to inhibit the RyR1 channel by clamping SPRY2 and α-solenoid 1 domains, thus reducing their mobility [6,20]. The physiological significance of the model. as discussed below, would be to allow FKBP12 to finetune the Ca^2+^ release from the SR over a wider range of activity than would be possible, with either an increase or decrease in RyR1 activity as appropriate in response to changes in the cellular environment.

As suggested for the biphasic effect of FKBP12 binding RyR2 [13], the most obvious alternative hypothesis to account for the results is that FKBP12 can bind to two sites with different accessibility to, or affinity for, FKBP12. However, only one binding site has been identified [4,5,6,20,26]. Another possibility is that the biphasic changes reflect FKBP12 binding and unbinding. However, the changes in gating parameters indicated separate mechanisms for activation and inhibition rather than binding and unbinding. In addition, S107 reduced activity to levels that were significantly less than the control, rather than control levels as might be expected with unbinding. It is also possible that FKBP12 interactions with RyR1 alter the binding of associated regulatory proteins and indirectly alter RyR1 activity. However, the biphasic changes were seen in solubilized channels, which lack associated proteins [16,17]. This list is not exhaustive, but it addresses major alternative possibilities.

### 4.4. RyR1 Association with FKBP12 in Muscle Fibers

FKBP12 binding to RyR1 is functionally important in intact fibers [3,34]. Our data suggest that channels in the muscle fibers are ~50% occupied by FKBP12, assuming the homogenate reflects in vivo FKBP12/RyR1 association. Given the ease of dissociation during processing, some FKBPs may be lost when fibers are disrupted, and the cellular RyR1 may be >50% occupied by FKBP12, although it is unlikely that occupation would be close to 100%. Submaximal occupation plus the ease of dissociation suggests a physiologically significant dynamic role for FKBP12 binding to RyR1 during changes in cellular conditions that modify RyR1 activity and SR Ca^2+^ release. While we can predict how FKBP12–RyR1 interactions may affect muscle function, there are numerous other FKBP12 targets in skeletal muscle fibers that can impact Ca^2+^ signaling. FKBP12 binds to TGFβR1 and reduces TGFβ signaling, broadly affecting tissue development and homeostasis [35]. Lee et al. [3] reported roles for FKBP12 in increasing SR Ca^2+^ uptake, increasing protein synthesis, slowing muscle fatigue, and up-regulating type I fibers. While enhanced FKBP12-depleted RyR1 channel activity contributes to muscle dysfunction [7,8], it can also have beneficial effects, including increased fatigue resistance [3]. The significance of reduced FKBP12 concentrations for RyR1 within the muscle fiber can be enhanced by the low-affinity association of FKBP12 with RyR1. Cellular [FKBP12]s as low as 0.2 µM are possible [8] and are in a concentration range where we find that inhibition is less dominant and activation is more apparent (Figure 3, Figure 4, Figure 5, Figure 6 and Figure 7). It was recently found that the embryonic deletion of FKBP12 causes early-onset and progressive Dilated Cardiac Myopathy, with increased cardiac oxidative stress, altered expression of proteins associated with cardiac remodeling and disease, and increased SR Ca^2+^ leaks [36]. It is possible that changes in embryonic [FKBP12] may also influence the development of skeletal muscle.

Low-affinity binding would facilitate dynamic changes in FKBP12 binding to RyR1 with changes in the cellular environment. Several factors that transiently change within the muscle fiber and alter RyR1 activity and FKBP12 binding to RyR1 include oxidizing and reducing conditions, S-nitrosylation, *S*-glutathionylation, and cytoplasmic [Ca^2+^] [8]. Cytoplasmic [Ca^2+^]-dependent effects on the changes in *P_o_* associated with FKBP12 binding are shown in Figure 3 and have been reported previously [9,11]. Together, these observations highlight the possibility that environmental changes could significantly alter FKBP12 binding to RyR1 in muscle fibers and add to the direct effects of each factor on RyR1 activity and Ca^2+^ release from the SR.

The effects of bilayer solution composition may contribute to the different reported effects of adding FKBP12 to RyR1. Notably, the compositions of the bilayer solutions that we used were similar to those used by Mei et al. [10], and our results were very similar in that the RyR1 *P_o_* following the addition of 5 µM of FKBP12 seldom fell below control levels within the lifetime of the bilayer, unless FKBP12 was experimentally stripped and *P_o_* fell further when S107 was also present. The effect of ionic composition on FKBP12 binding and RyR1 response is not surprising in that differences in ionic composition could well alter RyR1 structure, particularly in the mobile corona region, which contains the FKBP12 binding site.

## 5. Conclusions

We provide novel evidence for the activation of RyR1 channels in lipid bilayers following the high-affinity binding of FKBP12 to RyR1 during brief exposure to 1 nM to 5 µM of FKBP12, followed by inhibition with lower-affinity binding after 15 to 30 min of exposure to 0.2 to 5 µM of FKBP12. The results are consistent with a model of negative co-operativity in FKBP12 binding to four identical sites, one on each monomer of the RyR1 tetramer. We suggest that the structural consequences of FKBP12 binding to one or two non-adjacent subunits alter their orientation to allow greater mobility within the tetramer, increasing channel activity and, at the same time, reducing the affinity of unbound subunits for FKBP12. Lower-affinity FKBP12 binding to adjacent subunits with increasing [FKBP12] and longer exposure times stabilizes inter-subunit interactions, reducing mobility within the corona and reducing channel activity. We suggest that low-affinity binding can facilitate dynamic changes in FKBP12 binding to RyR1 with changes in the cellular environment.

## Figures and Tables

**Figure 2 cells-14-00157-f002:**
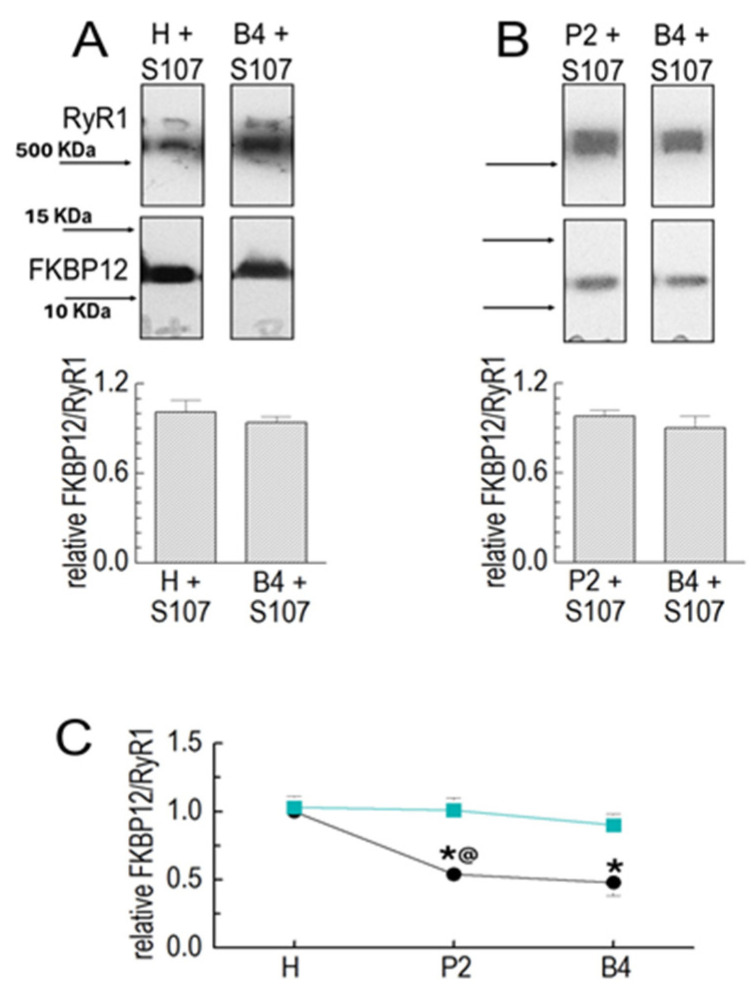
S107 prevents FKBP12’s dissociation from RyR1 during processing. Results for the Co-IP of RyR1 and FKBP12 in SR preparations in the following preparation in the presence of 20 µM of S107. Co-IP samples were loaded at protein concentrations of approximately 3, 6, and 9 μg. (**A**) RyR1 and FKBP12 in homogenate (H) and B4 vesicles in the presence of S107, as labeled. (**B**) RyR1 and FKBP12 in P2 and B4 vesicles in the presence of S107, as labeled. The images in each panel show bands from the same SDS-PAGE gel, and images in (**A**,**B**) are from the same gel as the images shown in Figure 1B,C, respectively. The upper image in each panel shows the RyR1 band, and the lower image shows the FKBP12 band, with vertically aligned images obtained from the same lane. Positions of MW markers are provided in (**A**) with corresponding arrows indicating marker positions in (**A,B**). The graphs (**A**) show the average relative FKBP12/RyR1 in the homogenate (**left**) and in B4 (**right**) processed with S107 (*n* = 5). The FKBP12/RyR1 ratios were first calculated for the RyR1 and FKBP12 bands in the same lanes; then, the H + S107 ratio was expressed relative to the H ratio (**left**), and the B4 + S107 ratio was expressed relative to the B4 ratio (**right**). The graphs in (**B**) show the average relative FKBP12/RyR1 ratio in P2 (**left**) and B4 (**right**) processed with S107 (*n* = 6). The FKBP12/RyR1 ratios were first calculated for the RyR1 and FKBP12 bands in the same lanes, and then, the ratio for P2 + S107 was expressed relative to the ratio in P2 alone, and the ratio for B4 + S107 was expressed relative to the ratio for B4 alone. The average values are indicated by broad vertical bars, and the s.e.m. is indicated by the vertical capped lines. The *n* values refer to the number of individual experiments. (**C**) A summary of results obtained with vesicle processing in the absence of S107 (black) and the presence (cyan) of S107. The asterisk * indicates a significant difference from the control, and the @ symbol indicates a significant difference from the S107 data for the P2 data.

**Figure 3 cells-14-00157-f003:**
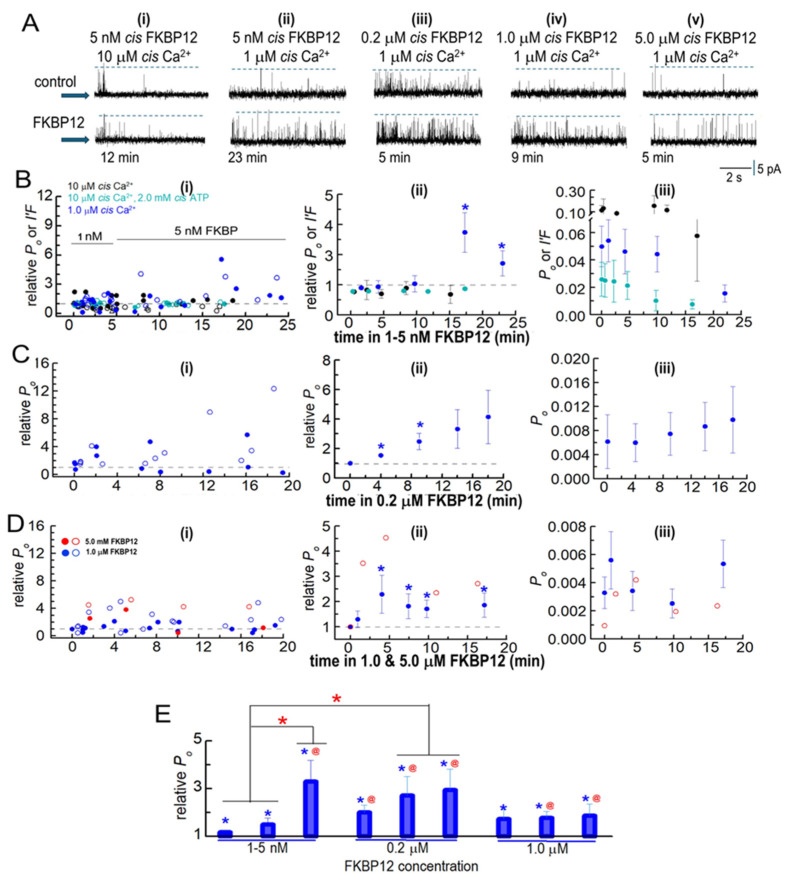
Biphasic FKBP12-induced changes in the activity of native RyR1 channels from B3 and B4 sucrose gradient fractions. RyR1 channel activity recorded in artificial lipid bilayers. (**A**) Current recordings obtained from native RyR1 channels during typical experiments over 4 to 8 min of control recording before FKBP12 was added to the cytoplasmic (cis) solution containing different [Ca^2+^] and [ATP] specified above the records. Upper “control” recordings were obtained before FKBP12 addition, and the lower recordings at the times below each record. The [FKBP12]s are provided at the top of each panel. Currents were at +40 mV, with upward deflections indicating channel opening from the baseline (indicated by the arrow on the left-hand side of each row) to the maximum open channel level (broken lines in each record). (**i**) 10 µM cis Ca^2+^: No change in activity after 12 min exposure to 5 nM FKBP12. (**ii**) 1 µM cis Ca^2+^: increased activity after 23 min with 5 nM FKBP12. (**iii**) 1 µM cis Ca^2+^: increased activity after 5 min with 0.2 µM FKBP12. (**iv**) 1 µM cis Ca^2+^: increased activity after 9 min with 1.0 µM FKBP12. (**v**) 1 µM cis Ca^2+^: increased activity after 5 min with 5.0 µM FKBP12. (**B**–**D**) Analysis of channel activity as a function of time and [FKBP12]. *P*_o_ was measured from single-channel recordings with 1 µM cis Ca^2+^ and *I*′*F* for multiple channel recordings with 10 µM cis Ca^2+^ and 10 µM cis Ca^2+^ plus 2 mM ATP in the cytoplasmic solutions. Data are plotted as a function of time after FKBP12 addition to the cytoplasmic solution. (**B**) 1–5 nM FKBP12 with 1 µM cis Ca^2+^ (royal blue symbols, *n* = 12), 10 µM cis Ca^2+^ (black symbols, *n* = 8), and 10 µM cis Ca^2+^ plus 2 mM ATP (cyan symbols, *n* = 12). (**C**) 0.2 µM FKBP12 (*n* = 6). (**D**) 1 µM FKBP12 (*n* = 10) and *n* = 2 for 5 µM FKBP12). The data for the one channel recorded with 5 µM FKBP12 are included because they provide additional evidence confirming the biphasic trends seen with 1 µM FKBP12. (**B**(**i**)–**D**(**i**)) Graphs showing individual measurements of relative *P*_o_ and relative *I*′*F* at +40 mV (filled circles) and at −40 mV (open circles) for all channels contributing to the average data in (**B**(**ii**)–**D**(**ii**)). (**B**(**ii**)–**D**(**ii**)) Graphs showing average combined relative *P*_o_ and relative *I*′*F* in (**Bii**) or average relative *P*_o_ only in (**C**(**ii**),**D**(**ii**)). All data in (**C**,**D**) were obtained from single-channel recordings with 1 µM cis Ca^2+^. (**B**(**iii**)–**D**(**iii**)) Graphs showing average *P*_o_ and average *I*′*F* in (**B**(**iii**)) or average *P*_o_ in (**C**(**iii**),**D**(**iii**)). In columns (**ii**,**iii**), filled symbols indicate the average values, with the s.e.m. indicated by vertical capped bars. (**E**) Data for each [FKBP12] shown in (**B**–**D**) have been rearranged into three time groups: 1–5 min, left bar; 7–10 min, middle bar; and 11–20 min, right bar. For 1–10 nM FKBP12, 1–5 min (*n* = 20); 7–10 min (*n* = 8); 11–20 min (*n* = 4). For 0.2 µM FKBP12, 1–5 min (*n* = 12); 7–10 min (*n* = 6); 11–20 min (*n* = 6). For 1.0 µM FKBP12, 1–5 min (*n* = 12); 7–10 min (*n* = 7); 11–20 min (*n* = 8). Blue asterisks * in (**B**–**E**) indicate significant differences from the control after the addition of FKBP12. The red asterisks (*) in (E) indicate significant differences from the same time period in 1–5 nM FKBP12. The red @ symbol in (**E**) indicates significant differences from 7 to 10 min in 1–5 nM FKBP12.

**Figure 4 cells-14-00157-f004:**
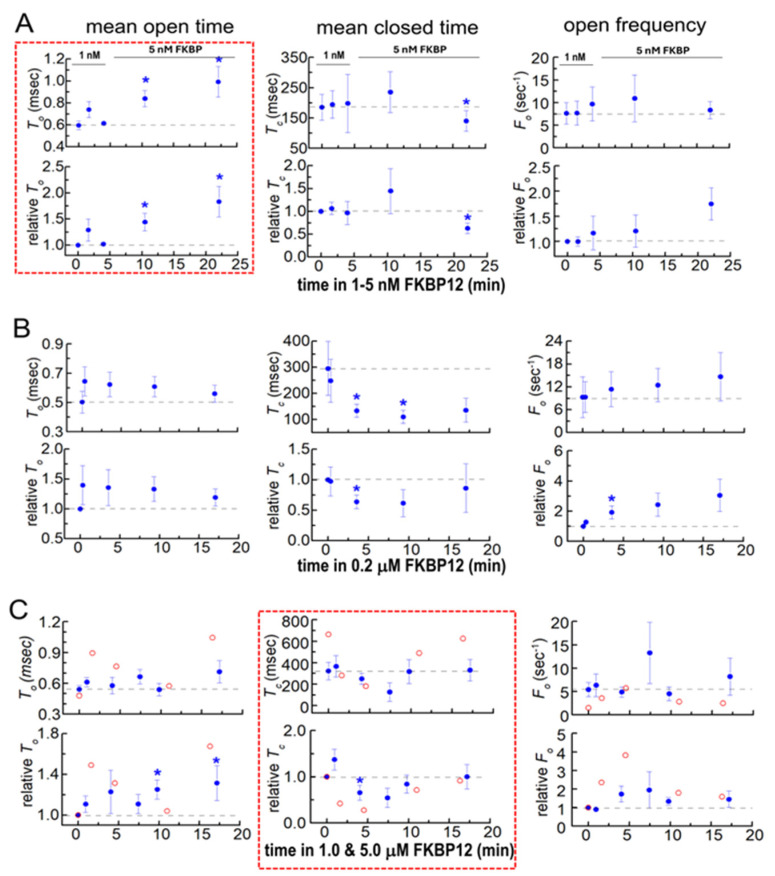
Changes in native RyR1 single-channel gating parameters during exposure to FKBP12. Gating parameters, mean open time (*T_o_*, left column), mean closed time (*T_c_*, middle column), and event frequency (*F_o_*, right column) were measured from the single-channel recording at the same time as the *P_o_* values in Figure 3. Refer to Figure 3 for general details that are not repeated here and for the number of experiments. Average data are plotted as a function of time and [FKBP12]. (**A**) 1–5 nM FKBP12; (**B**) 0.2 µM FKBP12; (**C**) 1 and 5 µM FKBP12. The data for the one channel recorded with 5 µM FKBP12 are included because they provide additional support for the trends seen with 1 µM FKBP12. Filled symbols indicate the average values, with the s.e.m. indicated by vertical capped bars. The broken red lines surround data sets that show trends in the gating parameter that are most clearly associated with the rapid onset of activation (*T_o_* in (**A**)). Blue asterisks * in indicate significant differences from the control after the addition of FKBP12.

**Figure 5 cells-14-00157-f005:**
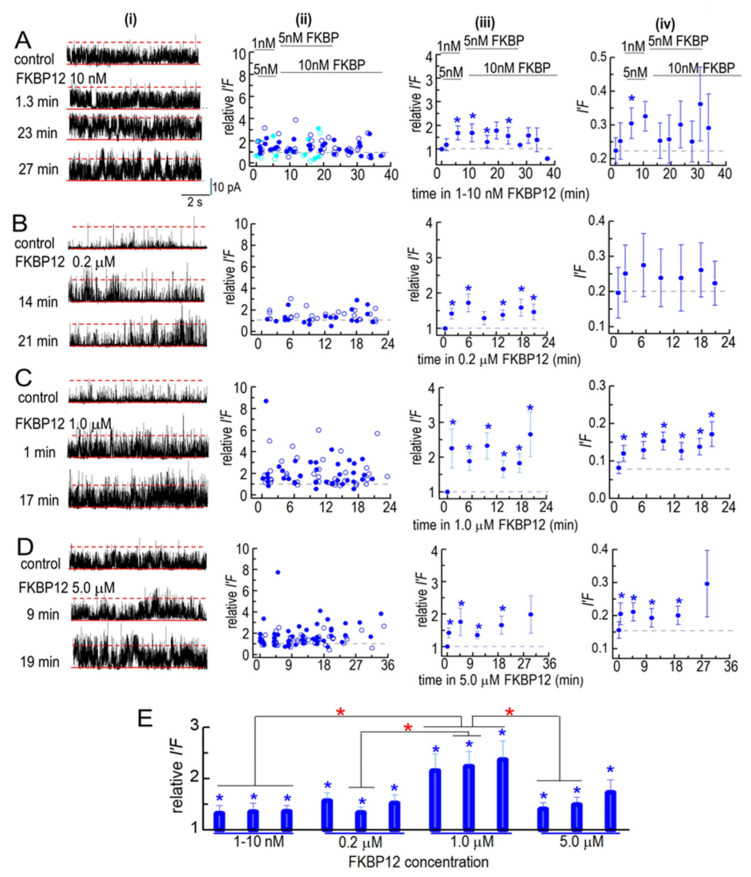
Effects of FBP12 on partially purified solubilized RyR1 channels. Ion channel activity recorded in artificial lipid bilayers. Current recordings obtained from solubilized RyR1 channels during typical experiments during 4 to 8 min control recording before FKBP12 was added to the cytoplasmic solution containing 1 µM free Ca^2+^ and for periods of ~20–38 min in the presence of FKBP12. (**A**–**D**) Column (**i**) shows examples of channel currents recorded before (upper record) and at the indicated times after FKBP12 addition (lower records). Currents are shown at +40 mV, with upward deflections indicating channel opening. In each recording, the solid red line marks the baseline (zero current level), and the broken red line marks the maximum open level for a single channel. The open level for a second full conductance opening is not specifically marked, as it is seldom clearly seen and cannot be separated from multiple sub-conductance openings. Column (**ii**): Plots of individual measurements of relative *I*′*F* at +40 mV (filled circles) and at −40 mV (open circles). Column (**iii**): average relative *I*′*F* (including independent values at +40 and −40 mV). Column (**iv**): Average *I*′*F*, shown to illustrate the approximate open probability of the solubilized channels (as described in the text). (**A**) In the initial 2 experiments, FKBP12 was added first at 1 nM and then increased to 5 nM (sky blue symbols). In the following 4 experiments, 5 nM FKBP12 was added first, and then, the concentration increased to 10 nM (royal blue symbols). The data from the two protocols were combined in the average effect of these low concentrations (*n* = 12). (**B**) Exposure to 0.2 µM FKBP12 (*n* = 8). (**C**) Exposure to 1.0 µM FKBP12 (*n* = 14). (**D**) Exposure to 5.0 µM FKBP12 (n = 16). (**E**) The data in (**A**–**D**) have been regrouped into three periods: For 1–10 nM FKBP12, 1–5 min (*n* = 24); 10–20 min (*n* = 23); 22–30 min (*n* = 27). For 0.2 µM FKBP12, 1–5 min (*n* = 16); 9–15 min (*n* = 16); 17–20 min (*n* = 16). For 1.0 µM FKBP12, 1–5 min (*n* = 28); 9–15 min (*n* = 28); 16–21 min (*n* = 20). For 5.0 µM FKBP12, 1–5 min (*n* = 32); 11–17 min (*n* = 28); 18–30 min (*n* = 18). The *n* values refer to the number of observations within each group. Blue asterisks (*****) in (**A**–**E**) indicate significant differences from the control after the addition of FKBP12. The red asterisks (*) in (**E**) indicate significant differences from the same time period in 1–5 nM FKBP12. For average data in (**A**,**D**), symbols show average values, and the s.e.m. is indicated by the vertical bars.

**Figure 6 cells-14-00157-f006:**
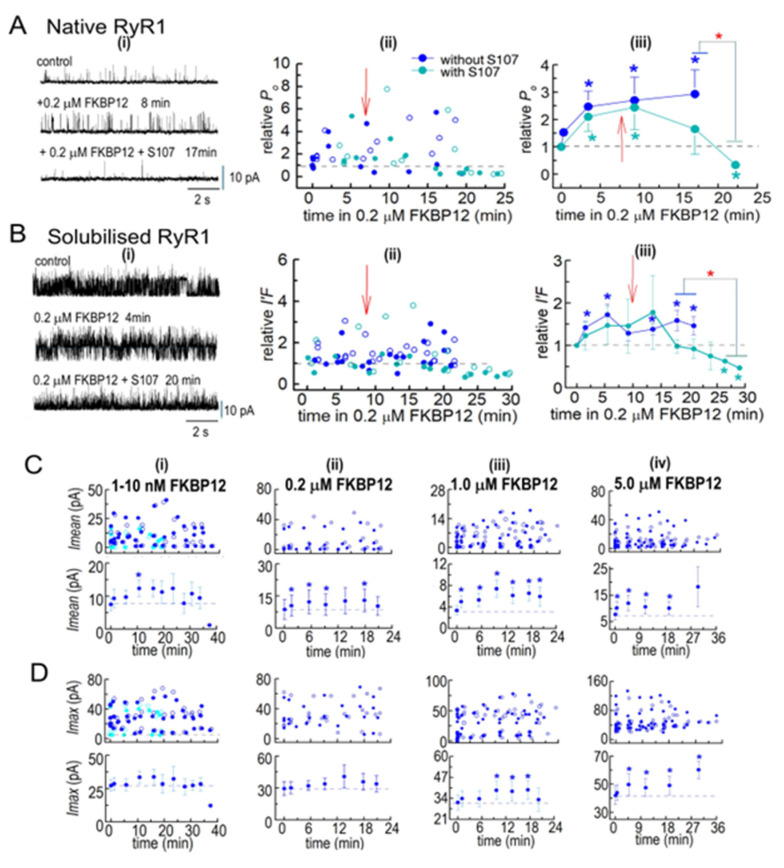
(**A**,**B**) FKBP12 inhibition of RyR1 is enhanced by S107. Ion channel activity recorded from native RyR1 channels (**A**(**i**)) and from solubilized RyR1 channels (**B**(**i**)), before FKBP12 addition to the cytoplasmic solution containing 1 µM free Ca^2+^ and then at various times over ~20–38 min exposures to 0.2 µM FKBP12 alone or 0.2 µM FKBP12 with 20 µM S107 added 5–10 min after FKBP12. Currents are shown at +40 mV, with upward deflections indicating channel opening. Relative *P_o_* for native RyR1 channels and *I*′*F* for solubilized RyR1 exposed to 0.2 µM FKBP12 in the absence of S107 in Figure 3 and Figure 5 are included in (**A**(**ii**),**B**(**ii**)), respectively, for comparison with the S107 data sets. Measurements of relative *P_o_* or relative *I*′*F* at +40 mV and −40 mV are included in the average relative *P_o_* and relative *I*′*F*, as described in the legends of Figure 3 and Figure 5. (**A**(**ii**),**B**(**ii**)) Individual measurements of relative *P_o_* or relative *I*′*F* at +40 mV (filled circles) and −40 mV (open circles) with 0.2 µM FKBP12 alone (royal blue symbols) or with 0.2 µM FKBP12 + S107 (cyan symbols). (**A**(**iii**),(**Biii**)) Graphs of average relative *P_o_* for native RyR1 channels (*n* = 6) and average *I*′*F* for solubilized RyR1 channels (*n* = 4), respectively, with 0.2 µM FKBP12 alone (royal blue symbols) or with 0.2 µM FKBP12 + S107 (cyan symbols). The red arrows in the graphs indicate approximate times of S107 addition to the cytoplasmic solution. (**C**,**D**) Effects of FKBP12 on current crossing the bilayer membrane, revealed in parameters used to calculate *I*′*F* for solubilized RyR1. The *Imean* data in (**C**) and *Imax* data in (**D**) were used to calculate *I*′*F* in Figure 5. Thus the number of experiments are the same as in Figure 5 and number of observations are provided in the legend to Figure 5. The upper graphs in each panel in (**C**,**D**) show individual *Imean* and *Imax* values, respectively, at +40 mV (filled circles) and −40 mV (open circles), while the lower graphs show average *Imean* and average *Imax*. (**C**(**i**),**D**(**i**)) 1–10 nM FKBP12: Sky blue symbols are from channels exposed to 1 nM and then 5 nM FKBP12, and royal blue symbols are from channels exposed to 5 nM and then 10 nM FKBP12; (**C**(**ii**),**D**(**ii**)) 0.2 µM FKBP12; (**C**(**iii**),**D**(**iii**)); 1.0 µM FKBP12; (**C**(**iv**),**D**(**iv**)) 5.0 µM FKBP12. The data from the 1–5 nM and 5–10 nM groups were combined to calculate the average effect of these low concentrations. For average data in (**A**–**D**), symbols indicate average values, and the s.e.m. is indicated by the vertical bars. The *n* values refer to the number of observations. The blue asterisks * indicate a significant difference from the control and the red asterisk * indicates a significant difference between the indicated data with S107 and without S107.

**Figure 7 cells-14-00157-f007:**
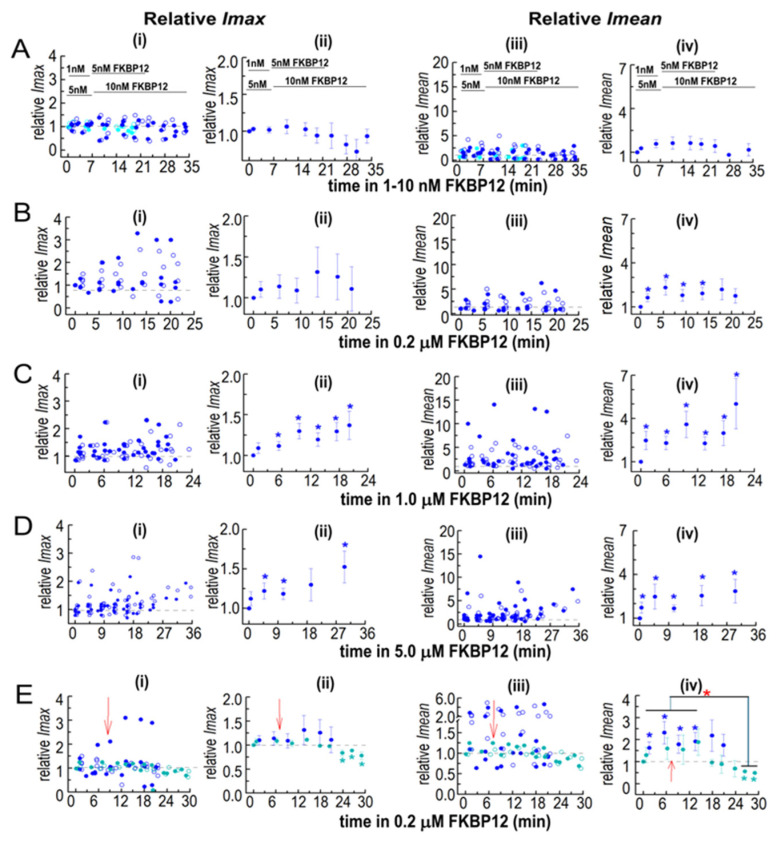
Normalization highlights the dominance of increases in *Imean* in the effects of FKBP12 on solubilized RyR1 channel activity and *I*′*F*. The relative *Imean* and *Imax* were calculated for the *Imean* and *Imax* data in Figure 5 and Figure 6. Both *Imean* and *Imax* in the presence of FKBP12 were normalized to their control value for each individual bilayer before obtaining the average relative values. Details of the experiments, including the number of observations, are the same as those in the legend in Figure 5. In (**A**–**E**), panel (**i**) shows the relative *Imax* in individual bilayers at +40 mV (filled circles) and −40 mV (open circles). Panel (**ii**) shows the average relative *Imax*. Panel (**iii**) shows relative *Imean* from individual bilayers at +40 mV (filled circles) and −40 mV (open circles). Panel (**iv**) shows the average relative *Imean*. For average data **i**n (**ii**,**iv**), symbols show average relative values, and the s.e.m. is indicated by the vertical capped bars. Data are shown in (**A**) for 1–10 nM FKBP12, in (**B**) for 0.2 µM FKBP12, in (**C**) for 1.0 µM FKBP12, and in (**D**) for 5.0 µM FKBP12, and in (**E**), data for 0.2 µM FKBP12 are repeated (filled royal blue circles) for comparison with 0.2 µM FKBP12 + S107 (cyan circles). The blue asterisks * indicate a significant difference between data with FKBP12 alone and control data. The cyan asterisks * indicate a significant difference between data with FKBP12 plus S107 and control data for those experiments and the red asterisk * indicates a significant difference between the indicated data with S107 and without S107.

**Figure 8 cells-14-00157-f008:**
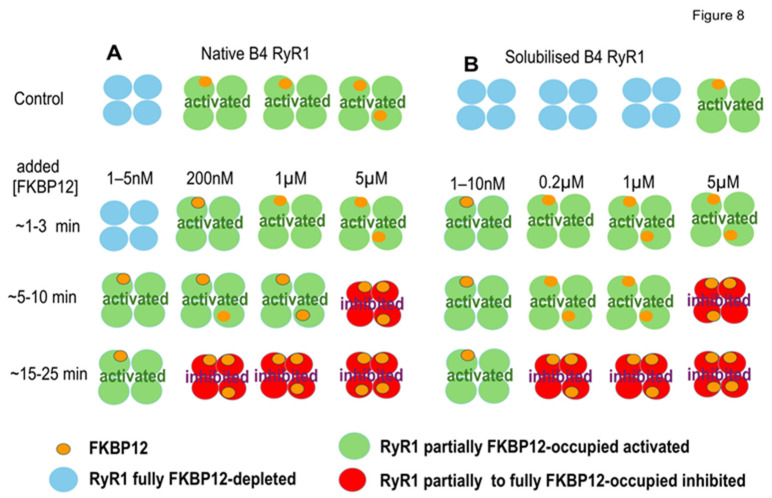
The negative co-operativity model suggests that changes in subunit occupation by FKBP12 within each tetramer might explain the changes in RyR1 channel activity associated with exogenous FKBP12 binding. (**A**,**B**) illustrate patterns of FKBP12 occupation of RyR1 subunits in RyR1 tetramers from B4 vesicles (**A**) and solubilized RyR1 (**B**). The control horizontal panel for B4 vesicles shows examples of four control RyR1 tetramers before adding exogenous FKBP12.From left to right, 1 tertamer with no FkBP12 bound, 2 tetramers with 1 FKBP12 bound, and 1 tetramer with 2 FKBP12 molecules bound, approximating the mean of 1.25 (Results). The control horizontal panel for four solubilized RyR1 tetramers is shown with 3 tetramers having 0 FKBP12 molecules bound, and 1 RyR1 tetramer with 1 FKBP bound. This is an overestimate of the 0.19 tetramers with FKBP12 bound (Discussion) but is shown to indicate that a small but finite number of solubilized RyR1 channels contain 1 subunit with FKBP12 bound. The remaining panels show increasing FKBP12 occupation with increasing time after exogenous FKBP12 addition and increasing [FKBP12]. The changes in occupation are suggested to explain the biphasic effects of FKBP12 on RyR1 channel activity presented in Figure 3, Figure 4, Figure 5, Figure 6 and Figure 7 and outlined in the Discussion. Note that the activated RyR1 tetramers are illustrated as having a larger cross-sectional profile as seen in the corona of activated RyRs, in contrast to the inhibited tetramers, which are depicted as smaller, representing the compact corona in a more closed conformation.

## Data Availability

Channel activity data were measured and analyzed using Channel 2 (developed by P. W. Gage and M. Smith, John Curtin School of Medical Research) or Channel 3 (developed by N. W. Laver, University of Newcastle). For more details on this software, please contact the corresponding author. The original Western blots for Figure 1 and Figure 2 are provided in the electronic Appendix A. All channel data are presented in the manuscript.

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
