# Peer review of "Complex Actions of FKBP12 on RyR1 Ion Channel Activity Consistent with Negative Co-Operativity in FKBP12 Binding to the RyR1 Tetramer"

_cells, 2025, doi:10.3390/cells14030157_

Round 1
Reviewer 1 Report
Comments and Suggestions for Authors
The manuscript by Richardson and colleagues uses western blots, co-immunoprecipitation and planar lipid bilayer experiments to explore the regulatory roles of FKBP12 binding to RyR1. This is of strong importance to the field, especially as this study provides a new model for FKBP12 regulation of RyR1.
Moderate concerns
· It is difficult to follow the sentences from lines 188 to 194. Especially, how the findings could suggest that FKBP12 has a lower binding affinity to RyR1 (relative to RyR2 binding). This suggestion is also complicated by the presence of FKBP12.6 in cardiac tissue. Please revise or remove this section of the Results.
· Figure 1 legend appears to have the description for panel H included in the description of panel F. Please correct.
· The statement on line 250-251 “The 82.5 ± 13% increase in cleaved FKBP12 binding was not 250 significantly different from the 75 ± 18% increase seen with GST-FBP12.6.” does not appear to align with the graphs shown in Figure 1. Please revise or confirm the data.
· Please add parameters (p value threshold, type, tail) of the statistical significance test to either section 2.7 or the figure legends.
Minor concerns:
· Remove unnecessary word “with” from line 61
· Remove unnecessary word “to” after “FKBP12” on line 316
Author Response
COMMENT 1: It is difficult to follow the sentences from lines 188 to 194. Especially, how the findings could suggest that FKBP12 has a lower binding affinity to RyR1 (relative to RyR2 binding). This suggestion is also complicated by the presence of FKBP12.6 in cardiac tissue. Please revise or remove this section of the Results.
RESPONSE TO COMMENT 1: Thank you for this comment. We agree that the section may have been difficult to follow and was speculative, particularly the final sentence. The text has been simplified and the final sentence removed. The speculation has been retained but with emphasis on speculation in the words “possibly indicating”. These changes can be found p4, final paragraph, lines 189-196 of the revised manuscript.
The original section was: “This result implies that most of the dissociation of FKBP12 from RyR1 occurs during processing from the homogenate to P2, rather than loss during movement though the sucrose gradient. In contrast to RyR1, ~70% of FKBP12/12.6 was lost from RyR2 with most dissociation occurring during centrifugation to obtain RyR2-enriched P4 vesicles [16]. Since processing to P2 was identical for both tissues, the greater loss of FKBP12 from RyR1 could suggest lower affinity binding of FKBP12 to RyR1. The greatest loss from RyR2 between P2 and P4 may be due to centrifugation as opposed to sucrose gradient rather than, or in addition to, RyR isoform differences.”
The revised section is as follows: “This result implies that most of the dissociation of FKBP12 from RyR1 occurs during the initial processing of the homogenate rather than during the following sucrose gradient to enrich the samples with vesicles containing RyR1. In contrast to RyR1, very little FKBP12 was lost from RyR2 during initial homogenate processing [13]. Since the processing techniques both skeletal and cardiac muscle homogenate were identical, FKBP12 was apparently more easily dissociated from RyR1 than from RyR2 at this stage of processing possibly suggesting a lower affinity binding of FKBP12 to RyR1 in the in the homogenate.”
- A change the response to a comment by Reviewer 2 in the preceding sentence in the revised manuscript (p4, line 186) is in blue font to distinguish the change in response to Reviewer 1
COMMENT 2: · Figure 1 legend appears to have the description for panel H included in the description of panel F. Please correct.
RESPONSE TO COMMENT 2: Thank you for this comment. The legend has been corrected in the revised manuscript and this change can be found in the figure legend at the top of p5, line 205 -208.
The original legend was:
Figure 1. FKBP12 dissociation from RyR1 during SR processing. (A). Sequential centrifugation and sucrose gradient fractionation in isolating SR vesicles enriched in RyR1 channels and solubil-ized RyR1. Relevant solute fractions are labelled S1and S5, relevant pellet labelled P2 and sucrose gradient bands labelled B1 to B4. (B-H). Co-IP of RyR1 and FKBP12 in SR preparations. Co-IP samples were loaded at protein concentrations of approximately 3, 6 and 9 μg. (B). RyR1 and FKBP12 in homogenate (H) and B4 vesicles (n = 5). (C). RyR1 and FKBP12 in P2 and B4 vesicles (n = 6). (D). RyR1 and FKBP12 in control incubated homogenate and homogenate incubated with GST-cleaved FKBP12 (cFKBP12) (n = 3). (E). RyR1 and FKBP12 in B4 and solubilized B4 (n = 5). (F). RyR1, GST-FKBP12, and FKBP12 in control incubated B4 and B4 incubated with cFKBP12 (n = 4). (G). Supernatant following Co-IP (n = 4). (B-H). The images in each figure show immune-stained bands from the same SDS-PAGE gel. The upper image in each panel shows the RyR1 band and the lower image shows the FKBP12 band. An additional band is shown for GST-GKBP12 in (F) and (G). All vertically aligned images in one panel were obtained from the same lane. MW markers are labelled in (B) and (F) and corresponding arrows indicating marker positions are shown in all panels. Unless otherwise stated, the graphs in each figure show the average relative FKBP/RyR2 ratios. The ratios were first calculated for RyR and FKBP bands in the same lanes and then the ratio for the right hand lane expressed relative to ratios for left hand “control” lane in the same blot. The average values are indicated by broad vertical bars and the s.e.m. indicated by the vertical capped lines. The n values refer to the number of individual experiments. The total FKBP12 density in (F) was calculated as (FKBP12 + GST-FKBP12/2).
The revised legend is as follows:
Figure 1. FKBP12 dissociation from RyR1 during SR processing. (A). Sequential centrifugation and sucrose gradient fractionation in isolating SR vesicles enriched in RyR1 channels and solubilized RyR1. Relevant solute fractions are labelled S1and S5, relevant pellet labelled P2 and sucrose gradient bands labelled B1 to B4. (B-H). Co-IP of RyR1 and FKBP12 in SR preparations. Co-IP samples were loaded at protein concentrations of approximately 3, 6 and 9 μg. (B). RyR1 and FKBP12 in homogenate (H) and B4 vesicles (n = 5). (C). RyR1 and FKBP12 in P2 and B4 vesicles (n = 6). (D). RyR1 and FKBP12 in control incubated homogenate and homogenate incubated with GST-cleaved FKBP12 (cFKBP12) (n = 3). (E). RyR1 and FKBP12 in B4 and solubilized B4 (n = 5). (F). RyR1, GST-FKBP12, and FKBP12 in control incubated B4 and B4 incubated with GST-FKBP12 (n = 4). (G). Supernatants following Co-IP, control incubated B4 and B4 incubated with GST-FKBP12 (n = 5). (H) B4 and B4 incubated with GST-cleaved FKBP12 (cFKBP12) (n = 3). (B-H). The images in each figure show immune-stained bands from the same SDS-PAGE gel. The upper image in each panel shows the RyR1 band and the lower image shows the FKBP12 band. An additional band is shown for GST-GKBP12 in (F) and (G). All vertically aligned images in one panel were obtained from the same lane. MW markers are labelled in (B) and (F) and corresponding arrows indicating marker positions are shown in all panels. Unless otherwise stated, the graphs in each figure show the average relative FKBP12/RyR1 ratios. The ratios were first calculated for RyR and FKBP bands in the same lanes and then the ratio for the right hand lane expressed relative to ratios for left hand “control” lane in the same blot. The average values are indicated by broad vertical bars and the s.e.m. indicated by the vertical capped lines. The n values refer to the number of individual experiments. The total FKBP12 density in (F) was calculated as (FKBP12 + GST-FKBP12/2).
COMMENT 3: The statement on line 250-251 “The 82.5 ± 13% increase in cleaved FKBP12 binding was not 250 significantly different from the 75 ± 18% increase seen with GST-FBP12.6.” does not appear to align with the graphs shown in Figure 1. Please revise or confirm the data.
RESPONSE TO COMMENT 3: Thank you for this comment. We accordingly corrected an error in the graph in Figure 1H. The mean and s.e.m given in the text (82.5 ± 13%) were correct. Now last sentence on p6.
COMMENT 4: Please add parameters (p value threshold, type, tail) of the statistical significance test to either section 2.7 or the figure legends.
RESPONSE TO COMMENT 4: The statistical parameters have been added to section 2.7, rather than the figure legends as they were essentially the same in all figures. The text has been modified as follows. P4, Line 168 – 176.
Data are presented as mean ± s.e.m. Significance was evaluated using the student’s t-test. For all Co-IP and channel data the threshold for significance was p < 0.05. For the Co-IP data, unless otherwise stated, the analysis was 2-tailed, the type was “type 1” (paired). For analysis of ion channel data, the analysis was either 1-tailed or 2-tailed and either type 1, type 2 or type 3 as appropriate. The effects of FKBP12 on channel activity were not voltage-dependent, so that measurements at +40 and -40 mV were included in the average data, as the two voltages provide independent measures the effects of FKBP12 at each concentration with current flow through the pore in opposite directions [13].
COMMENT 5: Remove unnecessary word “with” from line 61
RESPONSE TO COMMENT 5: Thank you. This has been done. Now line 60
COMMENT 6: Remove unnecessary word “to” after “FKBP12” on line 316
RESPONSE TO COMMENT 6: Thank you. This has been done. Now line 317
Reviewer 2 Report
Comments and Suggestions for Authors
Richardson and co-authors conducted a follow up study on the negative cooperativity among FKBP12 and RyR1 ion channels. Using a combination of co-immunoprecipitation and single channel recording measurements, the authors analyzed RyR1-FKBP12 binding in sarcoplasmic reticulum vesicles isolated from rabbit skeletal muscle. The authors found high affinity activation of RyR1 channels when FKBP12 binding sites on RyR1 tetramers were partially occupied and lower affinity inhibition of RyR1 channels with progressive FKBP12 occupation of RyR1 monomers. Other than performing another thorough editing of the manuscript (e.g., Line 55, extra "...or strong inhibition" at the end of the sentence), I do not have comments for improvement of the study/manuscript.
Author Response
We thank reviewer 2 for their comments and for correction to line 55.
“or strong inhibition” has been deleted from the end of the sentence on line 55
Reviewer 3 Report
Comments and Suggestions for Authors
This is an interesting and well-written paper describing the molecular mechanisms of FKBP12 regulation of RyR1. Attached are my comments and suggestions.

Author Response
COMMENT 1: The manuscript suggests there is no significant difference between the average ratio in the P2 and B4 fractions (Line 184), however Fig.1C indicates there is a significant difference. Bar graphs seem similar to me. Please clarify and make the corresponding modifications.
RESPONSE TO COMMENT 1: Thank you for this comment. The sentence as written was ambiguous. The wording has been revised and content divided into two sentences ion page 4, lines 185 to 189.
The original section was: In contrast to homogenate and B4, the amount of FKBP12 appears similar in the P2 and B4 bands and there is no significant difference between the average FKBP12/RyR1 ratio in the P2 and B4 fractions (Figure 1C and Figure 2C below).
NB upon checking there was in fact a small but significant difference between the average FKBP12/RyR1 ratio in the P2 and B4 fractions.
The revised section is as follows: In contrast to homogenate and B4, there is only a small although significant decline in average relative FKBP12/RyR1 in B4 compared to P2 (Figure 1C). The difference in the average FKBP12/RyR1 ratio between P2 and B4 is significantly less than the difference between homogenate and B4.
- Blue font has been used to distinguish between changes made in response to Reviewer 2 from changes in the following sentences made in response to comments Reviewer 1, which ae shown in red font.
COMMENT 2: · There are redactory errors in the text. A careful proof reading is needed, which I cannot do for the authors. Some examples include:
Line 63 line 74: depeleted.
Line 172: of is missing
Line 188: r is missing from though
RESPONSE TO COMMENT 2: Thank you for noticing these typographical errors. The errors have been corrected as follows.
Line 63 line 74: depeleted. “Depleted”. Corrected now shown on page 2, in line 62.
Line 172: of is missing. “measures of the effects”. Corrected on page 4, line 174.
Line 188: r is missing from though. The section has been rewritten in response to Reviewer 1, comment 1 – “through the sucrose gradient” has been deleted
The manuscript has undergone a careful proof read and several other typos have been corrected and minor changes made for clarification. These changes are shown in red font and noted in associated comments.